# Seismic Damage Identification of Composite Cable-Stayed Bridges Using Support Vector Machines and Wavelet Networks

**Zhongqi Shi** [1,2,3] , **Rumian Zhong** [1,2,3,*] **and Nan Jin** [1,2,3]

1   Key Laboratory of Urban Safety Risk Monitoring and Early Warning, Ministry of Emergency Management, Shenzhen 518055, China
2   Shenzhen Technology Institute of Urban Public Safety, Shenzhen 518000, China
3   National Science and Technology Institute of Urban Safety Development, Shenzhen 518000, China
*   Correspondence: zhongrumian@163.com

**Abstract:** A seismic damage identification method for composite cable-stayed bridges has been developed. The proposed method is based on a Support Vector Machine (SVM) and Wavelet Network (WN). A shaking table test of a composite cable-stayed bridge is employed to verify the identification accuracy of the WNSVM method; the test results show that the nonlinear Finite Element Model (FEM) can correctly simulate the single-tower cable-stayed bridge, and the learning samples of WNSVM can be produced based on the nonlinear FEM. The structural damage results identified by the WNSVM method are in good agreement with those obtained by the shaking table test, and the maximum error is less than 8%. Therefore, the WNSVM method can be used for the seismic damage identification of composite cable-stayed bridges.

**Keywords:** seismic damage identification; support vector machine; wavelet network; shaking table test; nonlinear finite element model; composite cable-stayed bridge





## 1. Introduction

A sophisticated highway system is supported by millions of bridges and viaducts in China. As time goes by, these bridges deteriorate under the action of typhoons, earthquakes, and overweight vehicle load, which seriously affects the safety of traffic infrastructure (Sohn et al., [1]; HE et al., 2013 [2]; Yeh et al., [3]; Pai & Sundaresan, [4]). Therefore, fast and accurate damage identification methods are very necessary, especially for bridges damaged under earthquake load. Fast damage identification and structural safety assessment methods are particularly critical to reducing the losses of the earthquake disaster.

In the field of structural health monitoring, a large number of scholars have explored the methods of damage identification, safety assessment, and structural reinforcement of bridge structures (Ou & Li, [5]; Chen & Wu, [6]; Xu et al., [7]; Duzgun et al., [8]; Richwalski et al., [9]). In terms of seismic damage identification, there are model-driven damage identification methods and data-driven damage identification methods (Lagaros, Tsompanakis, Psarropoulos & Georgopoulos, [10]; Vafaei, Adnan, Abd & Ahmad, [11]; Hou, Noori & Amand, [12]; Araújo & Laier, [13]; Chiu et al., [14–16]).

The theoretical background of model-based techniques can be found in studies by Doebling, Farrar, and Prime [17]; Providakis, Stefanaki, Voutetaki, Tsompanakis, and Stavroulaki [18]; Moyo, Brownjohn, Suresh, and Tjin, [19]; Pai, Young, and Lee, [20]. Application of these techniques to bridge structures has been introduced by Kim and Kawatni [21]; Park, Stubbs, Bolton, Choi, and Sikorsky [22]; Niu, Zong, and Chu [23]; Li, Bao, and Ou [24]; Li and Hao [25]; Bayissa, Haritos and Thelandersson [26]; Moaveni, He, Conte and De [27]; Ding, Li, Du, and Liu [28]. Some studies also use probability and multi-scale techniques for damage identification. Such as Oberkampf and Roy [29] proposed a Bayesian statistical method for damage identification. Zhong, Zong, Liu, and Zhou [30,31]

estimated the multi-scale model validation method of bridges based on structural health monitoring (SHM) system.

Compared with model-based techniques, data-driven methods need less computation and data storage, and they can identify the early structural damage online (Figueiredo, Figueiras, Park, Farrar & Worden, [32]). In recent years, a lot of papers have discussed data-driven methods. Reich [33] and Avci et al. [34] discussed the application of data-driven damage identification methods in civil engineering in detail. Yuan et al. [35] proposed an artificial neural network (ANN) which can be used for seismic estimation and damage assessment. Zhong et al. [36,37] and Yang et al. [38] systematically discussed the method of structural damage assessment of bridge columns under seismic load. Considering the influence of uncertainty, Duan et al. [39,40] proposed a group method of data handling (GMDH) neural network for parameter fitting. Because of their strong anti-interference and fast fitting ability, support vector machine has been widely used in civil engineering damage identification (González & Valdés, [41]). Jack and Nandi [42,43]; Samanta, Balushi, and Araimi [44] used statistical methods to preprocess the vibration signals as input features, and the SVMs method was presented to detect the structural fault of a rotating machine. Huang, Zhang, and Li [45] discussed an adaptive algorithm based on SVM to extract spatial features and classify high-resolution imagery. Sarp, Erener, Duzgun, and Sahin [46] used an image analysis and SVM method to demonstrate how efficiently they can be used for the automatic detection of buildings and changes in buildings after the Van-Ercis earthquake. Kim, Chong, Chong, and Kim [47] proposed an SVM method for damage identification of smart structures.

However, the parameters of the support vector machine (kernel parameters ($\sigma$), penalty factor (C), and insensitive loss function $\varepsilon$) directly affect the accuracy of its damage identification. Zhang [48]; Catbas, Gokce, and Frangopol [49] proposed a wavelet artificial neural network (WN) method which can be used to estimate the SVM parameters. Patil, Mandal, and Hegde [50] developed GA-SVM models to predict structural parameters. In our previous work [51], the WN was used to predict the network parameters.

This paper describes a seismic damage identification method for composite cable-stayed bridges in Section 2. A 1:30 scale model of a composite cable-stayed bridge is employed to verify the identification accuracy of the WNSVM method in Sections 3 and 4. Finally, Section 5 is the main conclusions.

## 2. Methodology

### 2.1. Damage Identification Method

Under the action of external load F, the dynamic equilibrium equation of a viscously damped system with mass [M], damping [C], and stiffness [K] can be written as

$$[M]\{\ddot{u}\} + [C]\{\dot{u}\} + [K]\{u\} = \{F\} \tag{1}$$

Moreover, if the response of the viscously damped system is $H$ and $\theta$ is the parameter of the system, $P$ is the external load [30]. This equation can be rewritten as,

$$H\theta = P \tag{2}$$

To obtain the parameter of the system $\theta$, Equation (2) can be expressed as

$$\theta = \left[H^T H\right]^{-1} H^T P \tag{3}$$

Since structural damage often only leads to changes in structural stiffness, the stiffness of a damaged structure can be calculated as

$$\theta_d = \left[H_d^T H_d\right]^{-1} H_d^T P \tag{4}$$

Based on Equations (3) and (4), the changes of unknown parameters with damage are likely to cause structural response vectors are different before and after damage. It can be shown as Equation (5):

$$\theta/\theta_d = \left[H^T H\right]^{-1} H^T P / \left[H_d^T H_d\right]^{-1} H_d^T P \tag{5}$$

As the structural response vector can be expressed as a 1-dimensional column vector, Equation (5) can be written as

$$\theta/\theta_d = h_d^2 H^T / h^2 H_d^T \tag{6}$$

where, $h = \left[H^T H\right]^{-1}$ and $h_d = \left[H_d^T H_d\right]^{-1}$ are the functions which are related to the structural response vector. $(\theta, H) \to (\theta_d, H_d)$ shows the change of structural response vectors.

As the structural response vectors of velocity $V$ and displacement $D$ are the integral function of acceleration $A$, Equation (6) can be written as

$$\theta/\theta_d = f(A)/f(A_d) \tag{7}$$

Based on Equation (7), the changes of unknown parameters $\theta/\theta_d$ are related to the structural response vectors of acceleration $A$ and $A_d$, it is shown as follows.

$$\theta/\theta_d = F(A, A_d) \tag{8}$$

However, the function of $F(A, A_d)$ is tedious and can be affected by many factors. It determines the difficulty of solving the function by mechanical method. In this paper, an intelligent fusion identification approach based on the Support Vector Machines (SVM) and Wavelet artificial neural Networks (WN) is presented and used to model a relationship between variables $\theta/\theta_d$ and independent variables $(A, A_d)$.

### 2.2. The Basic Principle of Support Vector Machines and Wavelet Networks

Under earthquake load, the structural responses corresponding to different structural parameters can be obtained through numerical simulation. In this section, the support vector machines and wavelet networks can be used to establish the functional relationship between independent variables $(A, A_d)$ and unknown parameters $\theta/\theta_d$. For the convenience of analysis, the independent variables $(A, A_d)$ can be represented as $x$, and unknown parameters can be represented as y. Then, the function $f$ is represented as

$$f(x) = \omega x + b \tag{9}$$

As $\varepsilon$ is the insensitive loss function, the optimized fitting function $f(x)$ can be obtained as

$$|y - f(x)| = \max\{0, |y - \omega x - b| - \varepsilon\} \tag{10}$$

Based on the Lagrangian dual transformation and least square method (Vapnik [52]), the optimization problem of $f(x)$ can be calculated as

$$\min_{\omega, \xi^{(*)}} \frac{1}{2} \sum_{m,n=1}^{k} (s_m^* - s_m)(s_n^* - s_n)(x_m \cdot x_n) + \varepsilon \sum_{m=1}^{k} (s_m^* + s_m) - \sum_{m=1}^{k} y_m(s_i^* - s_m) \tag{11}$$

where $s_m^*$ and $s_m$ are Lagrange multiplier. Then, the linear fitting function $f(x)$ can be formulated as

$$f(x) = \sum_{m=1}^{k} (s_m^* - s_m)(x_m \cdot x) + b \tag{12}$$

However, for the nonlinear fitting problem, it is often necessary to introduce a kernel function $k(x_i, x)$ to replace the $(x_i, x)$ in Equation (12), (Can, Xu & Xu, [53]), considering

of the complex and accidental in civil engineering, RBF kernel functions is selected in this paper.

$$k(x_m, x) = \exp\left(-|x_m - x|^2 / 2\sigma^2\right) \tag{13}$$

Finally, the optimal nonlinear fitting function $f(x)$ can be formulated as

$$f(x) = \sum_{m=1}^{k} (s_m^* - s_m) k(x_m \cdot x) + b \tag{14}$$

In order to obtain more accurate parameters of the support vector machine [51], this paper proposes to use a wavelet network to solve the problem by trial and least square method so as to improve the calculation efficiency of the support vector machine algorithm. As shown in Figure 1, it is a wavelet network with n input nodes and 1 output value, and the relationship between the inputs and output can be obtained as:

$$y = \sum_{m=1}^{n} q_k f_k = \sum_{m=1}^{n} f_k f\left(\frac{\sum_{i=1}^{l} w_{mi} x_i - b_m}{a_m}\right) \tag{15}$$

where $x_i$ and $y$ are represent inputs and output values, respectively. $a_k$ and $b_k$ is a fitting parameter of the wavelet network.

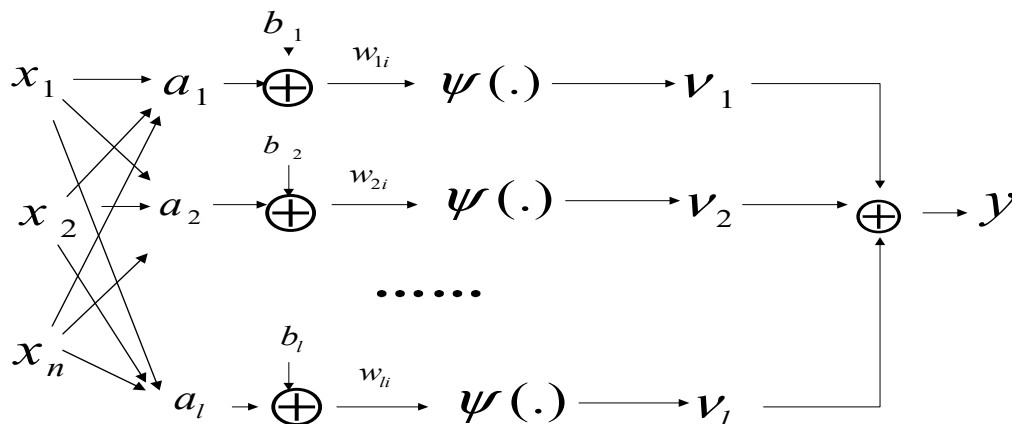

**Figure 1.** The wavelet neural network model.

### 2.3. The Framework of Damage Identification Based on WNSVM

The proposed damage identification approach is based on WNSVM, as shown in Figure 2. Firstly, a 1:30 scale model is tested on a shaking table, and seismic wave and bridge structure response are measured; Then, response accelerations of the structure are calculated based on FEM, and these response accelerations are selected as the samples for the SVM model; Thirdly, the parameters of SVM can be obtained by WN method; Finally, through the measured accelerations of the 1:30 scale composite cable-stayed bridge model, the damage values can be obtained. The identification results are also compared with the inspections of the damaged zones.

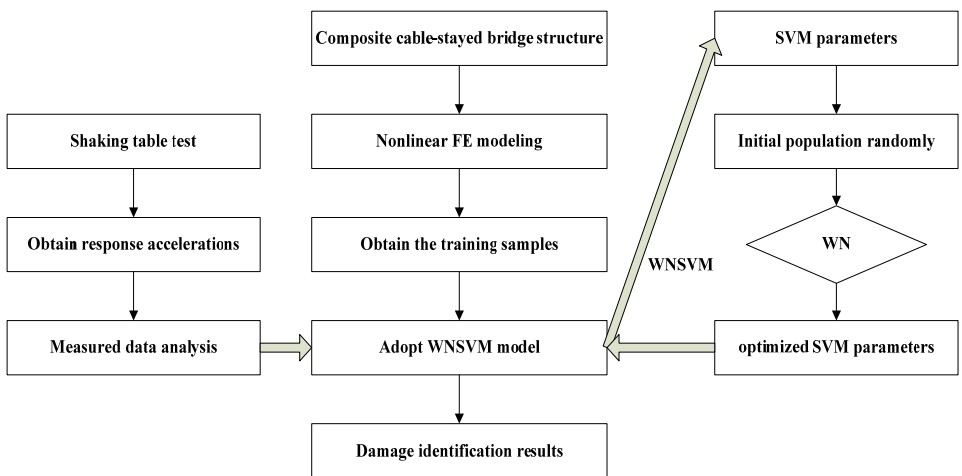

**Figure 2.** The framework of seismic damage identification is based on WNSVM.

## 3. Shaking Table Tests

*3.1. Experimental Description*

As shown in Figure 3a, the tested structure is a 1:30 scale model of the Guanhe bridge. Guanhe bridge is a 5-span continuous composite beam cable-stayed bridge, the main beam is an I-shaped steel concrete composite beam, in which the I-shaped steel longitudinal beam and cross beam are connected by high-strength bolts, and the prefabricated concrete deck is laid on the steel frame, forming a steel-concrete composite beam system.

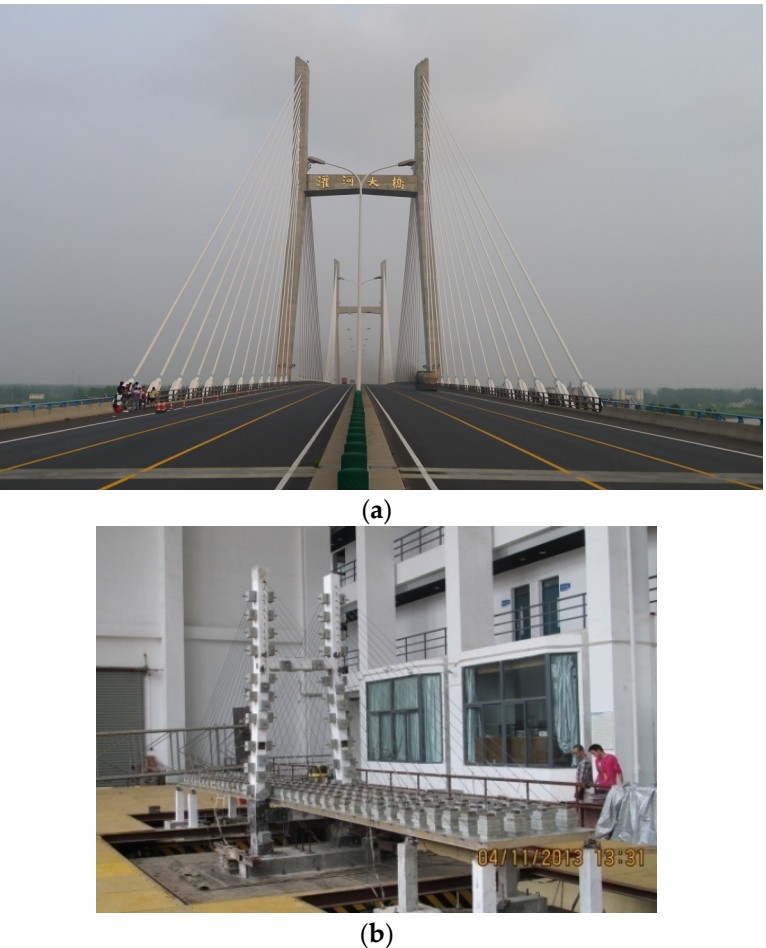

(**a**)

(**b**)

**Figure 3.** (**a**) Guanhe Bridge. (**b**) the single-tower cable-stayed bridge.

In Figure 3b, the long, wide, and high of the test single-tower cable-stayed bridge are 10.0 m, 2.0 m, and 4.2 m high. The main tower, auxiliary piers, and side piers of the model are reinforced concrete structures. The total height of the main tower of the model is 4.2 m; The main beam and cross beam of the composite beam is made of aluminum alloy plates after bending and assembling. The concrete bridge deck is made of aluminum alloy plates with a thickness of 1.5 mm. The total length of the composite beam is 10 m. The stay cable is made of high-strength prestressed steel wire with a diameter of 5 mm. There are 52 stay cables in the form of double cable planes. Two bilateral support are used to connect the main tower, main beam, and rubber plates are used to simulate two lateral limit devices; The auxiliary pier, side pier, and main beam are connected by bilateral support, of which the limit value of longitudinal bridge displacement is ± 50 mm, and the limit value of transverse bridge displacement is ±22.5 mm. The main tower, auxiliary pier, side pier, and vibration table are connected by bolts, main tower and bridge pier are respectively fixed on the three shaking tables. Limited by the ultimate bearing capacity of the shaking table and the model installation space, the final load counterweight of the main tower is 1160 kg, and the counterweight of the composite beam is 2518 kg.

As shown in Figure 4, The seismic simulation test of this model is carried out on a 6-DOF shaking table, and the structural responses under different seismic loads are collected through seven accelerometers, with a sampling frequency of 200 hz.

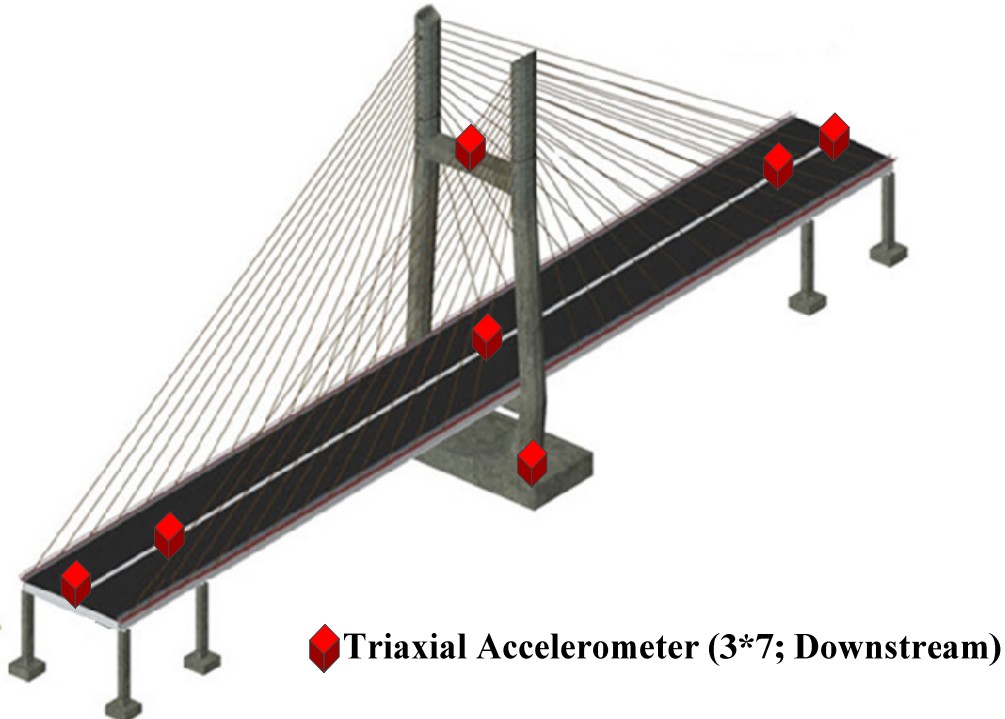

**Triaxial Accelerometer (3*7; Downstream)**

**Figure 4.** Sensors setting of single-tower cable-stayed bridge.

Table 1 shows earthquake cases, Peak Ground Acceleration in directions-X (longitudinal of the bridge) and Y (lateral of the bridge), and damage descriptions. In this paper, the Lander-amboy wave, CerroPrieto wave, El-centro wave, and Chi-chi wave are selected as seismic waves for the test. Their predominant periods are located in different ranges, respectively 0.15–0.20, 0.30–0.35, 0.50–0.60, 1.30–1.40 s, and peak ground acceleration of different seismic waves are also different. Through the test, the impact of seismic waves with different predominant periods and peak ground acceleration on the structure of a cable-stayed bridge can be investigated.

**Table 1.** Earthquake cases.

| Case | Seismic Waves | Peak Ground Acceleration (m/s$^2$) | | Damage Descriptions |
| --- | --- | --- | --- | --- |
| | | X-Direction | Y-Direction | |
| 1 | Lander-amboy | 0.5 | 0.394 | No obvious cracks |
| 2 | Cerro Prieto | 0.5 | 0.538 | No obvious cracks |
| 3 | Chi-chi | 0.5 | 0.495 | No obvious cracks |
| 4 | El-Centro | 0.5 | 0.308 | No obvious cracks |
| 5 | Cerro Prieto | 1 | 1.076 | Minor cracks in X-direction support |
| 6 | El-Centro | 2 | 1.23 | Lots of cracks in X-direction support |
| 7 | Chi-chi | 2 | 1.98 | Lots of cracks in X-direction support |
| 8 | El-Centro | 3 | 2.818 | Lots of cracks in X-direction support, Minor cracks in Y-direction support and stay cable anchor block |
| 9 | Chi-chi | 3 | 2.951 | Lots of cracks in X-direction and Y-direction support, Minor cracks in stay cable anchor block and bridge tower |
| 10 | Chi-chi | 4 | 5.981 | lots of major cracks appeared |

As shown in Table 1 and Figure 5, when the peak horizontal acceleration reaches 0.1 g, a small number of cracks begin to appear on the x-direction supports. With the continuous increase in the peak acceleration, the cracks tend to expand rapidly. When subjected to super-strong earthquake excitations (0.4 g Chi-chi), lots of cracks appeared on the surfaces of supports, stay cable anchor block, and bridge tower.

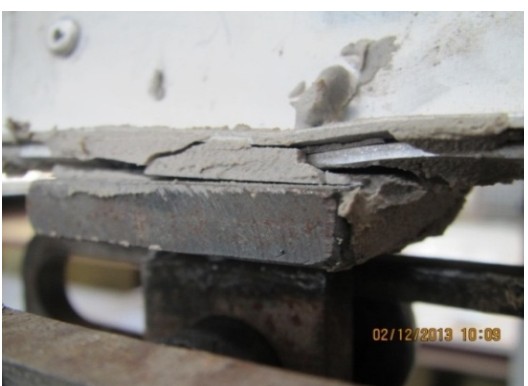 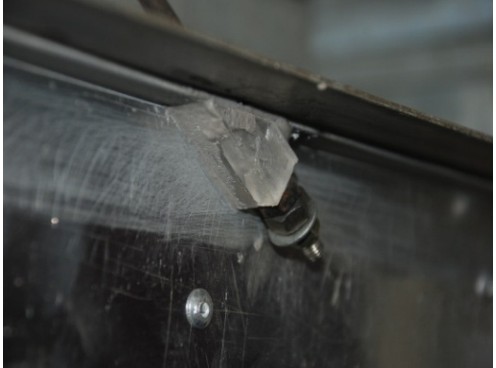

(**a**) cracks in the Supports       (**b**) cracks in stay cable anchor block

**Figure 5.** Visual inspection of damage.

### 3.2. Nonlinear FEM Simulations

Based on Newmark Constant Average Acceleration method (Chopra, [54]) and Modified Takeda Hysteretic Rule (Sake, [55]), the nonlinear FEM is shown in Figure 6. The finite element model of single tower cable-stayed bridge based on implicit integration is established by using large-scale general software ANSYS. The main tower is simulated by the Solid65 element, the longitudinal reinforcement of the main tower is simulated by the Link10 element, and the concrete material is simulated by using the Mander concrete model considering the effect of the stirrup. The main beam, cross beam, and small longitudinal beam of the bridge deck are simulated by Beam188 element, the bridge deck is simulated by Shell63 element, and the connection between the bridge deck and the composite beam is simulated by node coupling, the bridge deck counterweight is simulated by Mass21 mass element. The cable is simulated by the Link10 element. The connection between the cable and the main concrete tower and the composite beam is simulated by node coupling. In the finite element model, the bearing is simulated by the nonlinear spring element Combin39. The stiffness of the bearing is determined by the measured dynamic characteristics of

the model. At the same time, the displacement limit of the bearing is the same as the actual model.

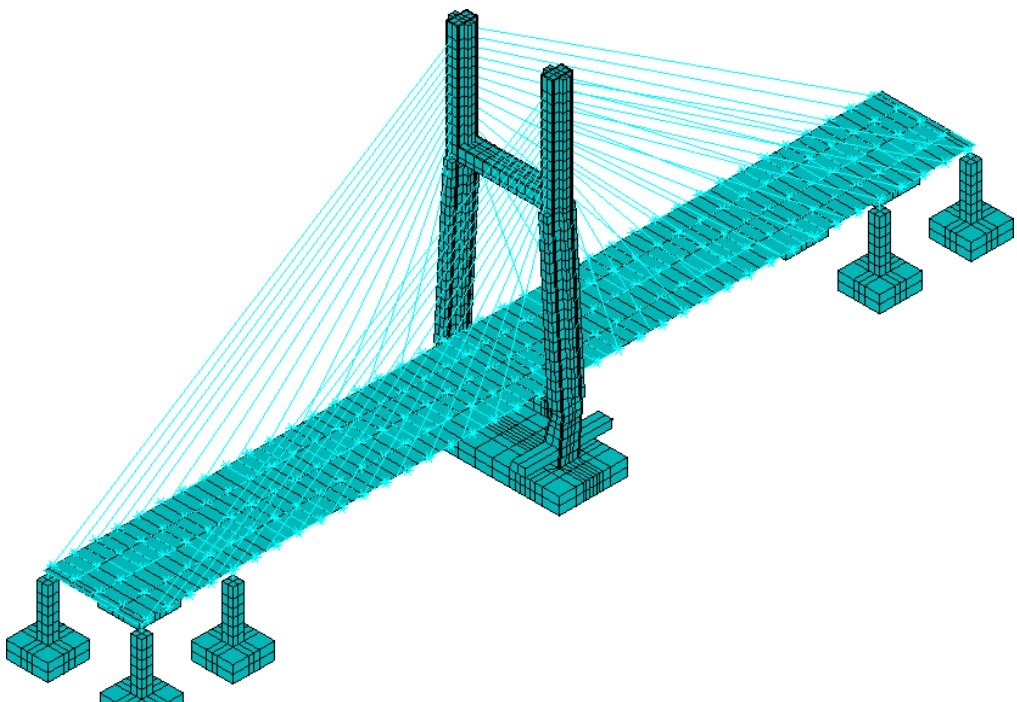

**Figure 6.** The nonlinear FEM of single-tower cable-stayed bridge.

Then, based on the nonlinear FEM, the structure responses of single-tower cable-stayed bridge under earthquake excitations (in Table 1) can be calculated, and these calculated values are compared with the measured values. The results of cases 1 and 9 are shown in Figure 7. Other cases have been introduced in the literature (Huang, Zong, Li & Xia, [56]). From there, it can be seen that the nonlinear FEM can correctly simulate the single-tower cable-stayed bridge model, and the learning samples of the WNSVM model in this study can be produced based on the nonlinear FEM.

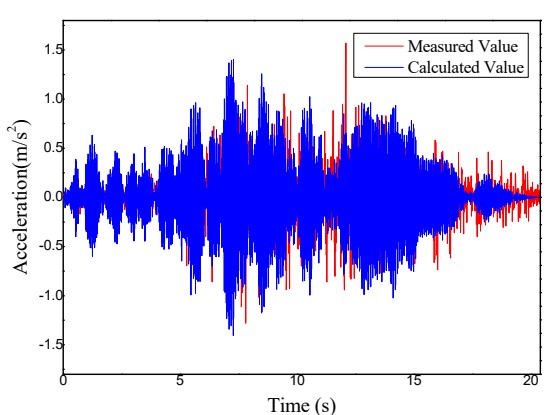

(**a**) X-direction acceleration of bridge tower under case 1

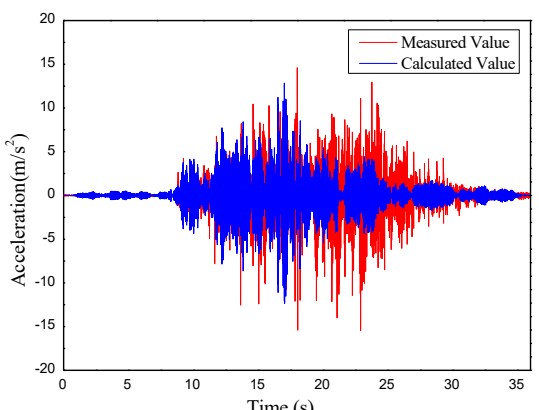

(**b**) X-direction acceleration of bridge tower under case 9

**Figure 7.** *Cont.*

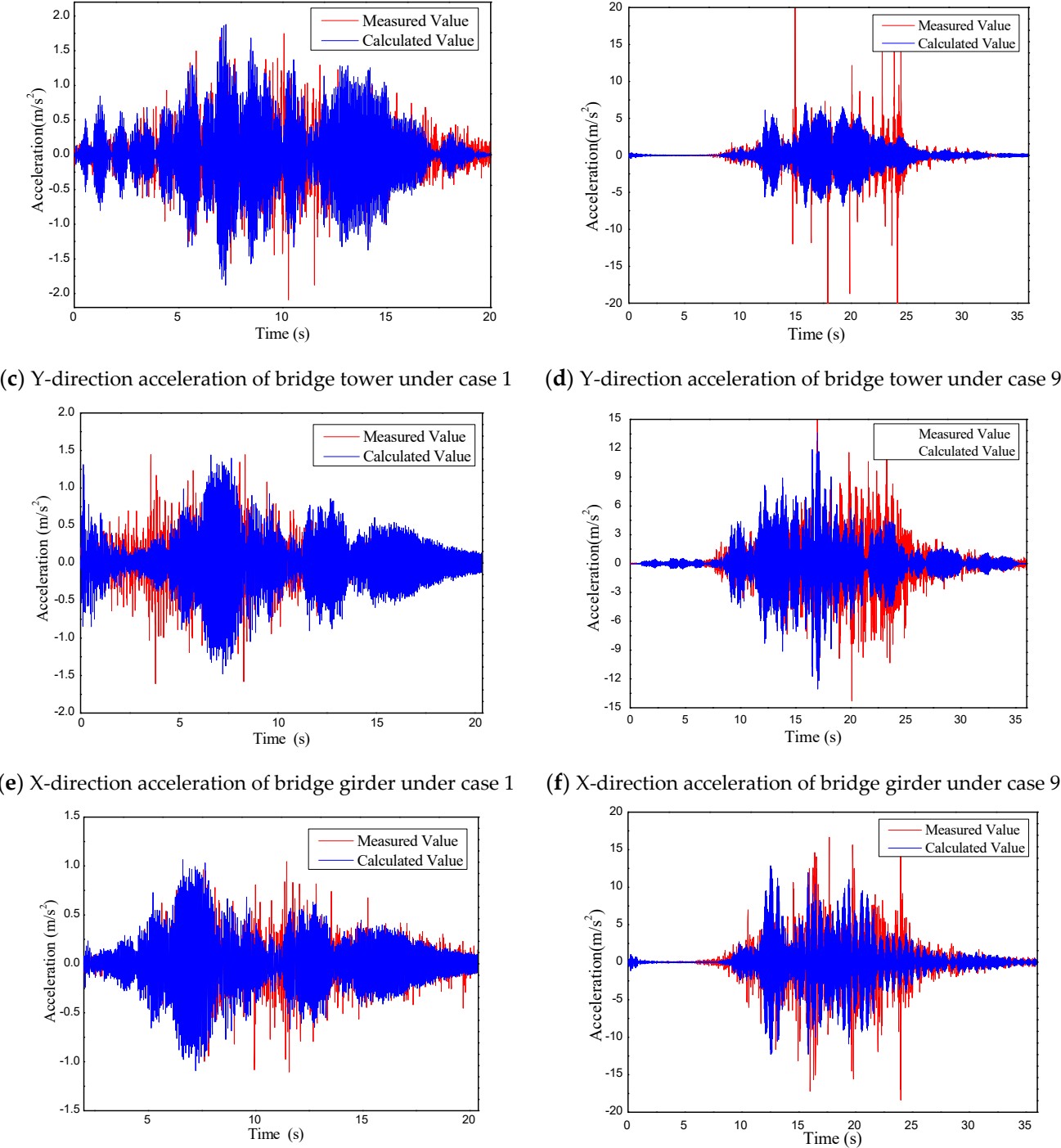

(**c**) Y-direction acceleration of bridge tower under case 1　(**d**) Y-direction acceleration of bridge tower under case 9

(**e**) X-direction acceleration of bridge girder under case 1　(**f**) X-direction acceleration of bridge girder under case 9

(**g**) Y-direction acceleration of bridge girder under case 1　(**h**) Y-direction acceleration of bridge girder under case 9

**Figure 7.** Comparison of acceleration time history curve of the cable-stayed bridge between test and FEA.

## 4. Damage Identification Using WNSVM

### 4.1. Damage Parameters Selection

Based on the shaking table tests in Section 3 and many pieces of literature (Lautour & Omenzetter, [57]; Ni, Zhou & Ko, [58]; Mohammadreza, Azlan, Ahmad & Abd, [59]) about bridges damaged in the earthquake, the stiffness of supports in directions-X and Y, the stiffness of bridge tower are selected as the damage parameters in this paper, as shown in Table 2. The initial values of these parameters can be obtained based on the ambient

vibration testing and model updating method (Huang, Zong, Li & Xia, [56]). On account of the damage of stay cable can be detected based on the measured cable force directly; it is not the damage parameters in this paper.

**Table 2.** Damage Parameters Selection.

| Structural Stiffness | k1, k2, k3 (N/m) | k4, k5, k6 (N/m) | k7 (N/m) | k8 (N/m) |
| --- | --- | --- | --- | --- |
| | Stiffness of Support in x Direction | Stiffness of Support in y Direction | Stiffness of Bridge Tower in y Direction | Stiffness of Bridge Tower in x Direction |
| initial values | $2 \times 10^6$ | $2.5 \times 10^7$ | $9.32 \times 10^5$ | $2.39 \times 10^6$ |

### 4.2. Training and Testing of WNSVM for Damage Identification

In this paper, damage index $D$ was employed for Damage quantification. It can be calculated from

$$D = K_d/K \tag{16}$$

where $K$ and $K_d$ are the damage parameters before and after damage, respectively. Then, Based on the D-optimal design method, as $0.05 < D < 1.5$, a 166-sample data set was calculated, and $0.05 < D < 1.5$, 163 samples were used to train the WNSVM, and the other three samples were used to verify the WNSVM, as shown in Tables 3 and 4. ground motion intensity and measured acceleration responses are inputs, and the structural parameters are outputs. Substituting these samples into the nonlinear FEM, acceleration responses of single-tower cable-stayed bridge under each case of earthquake excitations (in Table 3) can be calculated, and the mean value of the accelerations is used as the representative values in this paper. These training and testing samples were inputted into the WNSVM framework, which had introduced in Section 2. The SVM parameters also could be obtained based on the WN method, kernel function parameter $\sigma = 23.5$, regularization parameter C = 96.2, and regression approximation error control parameter $\varepsilon = 0.0138$. Table 5 and Figure 8 show the damage identification ability of the WNSVM method. It shows that the WNSVM method can be used to the seismic damage identification of Composite Cable-Stayed Bridge, and the maximum error is less than 8%.

**Table 3.** Training samples.

| Samples | | | 1 | 2 | 3 | ... | 161 | 162 | 163 |
| --- | --- | --- | --- | --- | --- | --- | --- | --- | --- |
| | Case | | 0.05 g Lander-Amboy | 0.05 g Cerro Prieto | 0.05 g EL-Centro | | 0.05 g Cerro Prieto | 0.05 g EL-Centro | 0.05 g Chi-Chi |
| K(N/m) | Support | $k_1$ | $2.0 \times 10^7$ | $2.0 \times 10^7$ | $2.0 \times 10^7$ | ... | $2.0 \times 10^7$ | $2.0 \times 10^7$ | $2.0 \times 10^7$ |
| | | $k_2$ | $2.0 \times 10^7$ | $2.0 \times 10^7$ | $2.0 \times 10^7$ | ... | $2.0 \times 10^7$ | $2.0 \times 10^7$ | $2.0 \times 10^7$ |
| | | $k_3$ | $2.0 \times 10^7$ | $2.0 \times 10^7$ | $2.0 \times 10^7$ | ... | $2.0 \times 10^7$ | $2.0 \times 10^7$ | $2.0 \times 10^7$ |
| | | $k_4$ | $2.5 \times 10^8$ | $2.5 \times 10^8$ | $2.5 \times 10^8$ | ... | $2.5 \times 10^8$ | $2.5 \times 10^8$ | $2.5 \times 10^8$ |
| | | $k_5$ | $2.5 \times 10^8$ | $2.5 \times 10^8$ | $2.5 \times 10^8$ | ... | $2.5 \times 10^8$ | $2.5 \times 10^8$ | $2.5 \times 10^8$ |
| | | $k_6$ | $2.5 \times 10^8$ | $2.5 \times 10^8$ | $2.5 \times 10^8$ | ... | $2.5 \times 10^8$ | $2.5 \times 10^8$ | $2.5 \times 10^8$ |
| | Bridge tower | $k_7$ | $9.3 \times 10^6$ | $9.3 \times 10^6$ | $9.3 \times 10^6$ | ... | $9.3 \times 10^6$ | $9.3 \times 10^6$ | $9.3 \times 10^6$ |
| | | $k_8$ | $2.4 \times 10^7$ | $2.4 \times 10^7$ | $2.4 \times 10^7$ | ... | $2.4 \times 10^7$ | $2.4 \times 10^7$ | $2.4 \times 10^7$ |
| $K_d$(N/m) | Support | $k_{d1}$ | $2.4 \times 10^7$ | $1.0 \times 10^7$ | $2.4 \times 10^7$ | ... | $1.3 \times 10^7$ | $2.4 \times 10^7$ | $2.4 \times 10^7$ |
| | | $k_{d2}$ | $2.0 \times 10^6$ | $2.4 \times 10^7$ | $2.4 \times 10^7$ | ... | $2.0 \times 10^7$ | $2.4 \times 10^7$ | $2.0 \times 10^6$ |
| | | $k_{d3}$ | $1.3 \times 10^7$ | $2.0 \times 10^6$ | $2.4 \times 10^7$ | ... | $2.4 \times 10^7$ | $2.4 \times 10^7$ | $2.0 \times 10^6$ |
| | | $k_{d4}$ | $3.0 \times 10^8$ | $2.5 \times 10^7$ | $3.0 \times 10^8$ | ... | $2.5 \times 10^7$ | $2.5 \times 10^7$ | $2.5 \times 10^7$ |
| | | $k_{d5}$ | $2.5 \times 10^7$ | $3.0 \times 10^8$ | $2.5 \times 10^7$ | ... | $6.5 \times 10^7$ | $3.0 \times 10^8$ | $3.0 \times 10^8$ |
| | | $k_{d6}$ | $2.5 \times 10^7$ | $3.0 \times 10^8$ | $1.0 \times 10^8$ | ... | $1.4 \times 10^7$ | $1.2 \times 10^8$ | $1.9 \times 10^8$ |
| | Bridge tower | $k_{d7}$ | $9.3 \times 10^5$ | $1.1 \times 10^7$ | $9.4 \times 10^6$ | ... | $1.1 \times 10^7$ | $1.1 \times 10^7$ | $9.3 \times 10^5$ |
| | | $k_{d8}$ | $2.9 \times 10^7$ | $6.3 \times 10^6$ | $1.7 \times 10^7$ | ... | $1.1 \times 10^7$ | $2.8 \times 10^7$ | $2.4 \times 10^6$ |

**Table 3.** *Cont.*

| Samples | | | 1 | 2 | 3 | ... | 161 | 162 | 163 |
|---------|---|---|---|---|---|-----|-----|-----|-----|
| Case | | | 0.05 g Lander-Amboy | 0.05 g Cerro Prieto | 0.05 g EL-Centro | | 0.05 g Cerro Prieto | 0.05 g EL-Centro | 0.05 g Chi-Chi |
| $A$ (m/s$^2$) | Support | $a_1$ | 0.43 | 0.34 | 0.31 | ... | 0.34 | 0.31 | 0.40 |
| | | $a_2$ | 0.43 | 0.34 | 0.31 | ... | 0.34 | 0.31 | 0.40 |
| | | $a_3$ | 0.32 | 0.25 | 0.23 | ... | 0.25 | 0.23 | 0.30 |
| | | $a_4$ | 0.21 | 0.26 | 0.08 | ... | 0.26 | 0.08 | 0.11 |
| | | $a_5$ | 0.21 | 0.26 | 0.08 | ... | 0.26 | 0.08 | 0.11 |
| | | $a_6$ | 0.18 | 0.22 | 0.07 | ... | 0.22 | 0.07 | 0.10 |
| | Bridge tower | $a_7$ | 0.62 | 0.48 | 0.45 | ... | 0.48 | 0.45 | 0.58 |
| | | $a_8$ | 0.98 | 1.21 | 0.38 | ... | 1.21 | 0.38 | 0.52 |
| $A_d$(m/s$^2$) | Support | $a_{d1}$ | 0.40 | 0.52 | 0.19 | ... | 0.50 | 0.21 | 0.27 |
| | | $a_{d2}$ | 2.24 | 0.26 | 0.24 | ... | 0.34 | 0.23 | 1.71 |
| | | $a_{d3}$ | 0.40 | 1.08 | 0.22 | ... | 0.74 | 0.24 | 1.51 |
| | | $a_{d4}$ | 0.16 | 0.83 | 0.06 | ... | 0.83 | 0.43 | 0.35 |
| | | $a_{d5}$ | 0.76 | 0.26 | 0.35 | ... | 0.22 | 0.08 | 0.08 |
| | | $a_{d6}$ | 0.92 | 0.14 | 0.07 | ... | 0.24 | 0.12 | 0.10 |
| | Bridge tower | $a_{d7}$ | 2.85 | 0.36 | 0.45 | ... | 0.36 | 0.34 | 3.62 |
| | | $a_{d8}$ | 0.89 | 3.75 | 0.39 | ... | 3.19 | 0.39 | 1.62 |

**Table 4.** Testing samples.

| Samples | 1 | 2 | 3 | Samples | 1 | 2 | 3 | Samples | 1 | 2 | 3 |
|---------|---|---|---|---------|---|---|---|---------|---|---|---|
| $k_1$ | $2.0 \times 10^7$ | $2.0 \times 10^7$ | $2.0 \times 10^7$ | $a_1$ | 0.31 | 0.34 | 0.40 | $a_{d1}$ | 0.19 | 0.52 | 0.27 |
| $k_2$ | $2.0 \times 10^7$ | $2.0 \times 10^7$ | $2.0 \times 10^7$ | $a_2$ | 0.31 | 0.34 | 0.40 | $a_{d2}$ | 0.24 | 0.26 | 1.71 |
| $k_3$ | $2.0 \times 10^7$ | $2.0 \times 10^7$ | $2.0 \times 10^7$ | $a_3$ | 0.23 | 0.25 | 0.30 | $a_{d3}$ | 0.22 | 1.08 | 1.51 |
| $k_4$ | $2.5 \times 10^8$ | $2.5 \times 10^8$ | $2.5 \times 10^8$ | $a_4$ | 0.08 | 0.26 | 0.11 | $a_{d4}$ | 0.06 | 0.83 | 0.35 |
| $k_5$ | $2.5 \times 10^8$ | $2.5 \times 10^8$ | $2.5 \times 10^8$ | $a_5$ | 0.08 | 0.26 | 0.11 | $a_{d5}$ | 0.35 | 0.26 | 0.08 |
| $k_6$ | $2.5 \times 10^8$ | $2.5 \times 10^8$ | $2.5 \times 10^8$ | $a_6$ | 0.07 | 0.22 | 0.10 | $a_{d6}$ | 0.07 | 0.14 | 0.10 |
| $k_7$ | $9.3 \times 10^6$ | $9.3 \times 10^6$ | $9.3 \times 10^6$ | $a_7$ | 0.45 | 0.48 | 0.58 | $a_{d7}$ | 0.45 | 0.36 | 3.62 |
| $k_8$ | $2.4 \times 10^7$ | $2.4 \times 10^7$ | $2.4 \times 10^7$ | $a_8$ | 0.38 | 1.21 | 0.52 | $a_{d8}$ | 0.39 | 3.75 | 1.62 |

**Table 5.** Testing results.

| Samples | 1 | | | 2 | | | 3 | | |
|---------|---|---|---|---|---|---|---|---|---|
| | Reference Value | Calculated Value | Error (%) | Reference Value | Calculated Value | Error (%) | Reference Value | Calculated Value | Error (%) |
| $k_{d1}$ | $2.4 \times 10^7$ | $2.5 \times 10^7$ | 3.52 | $1.0 \times 10^7$ | $1.0 \times 10^7$ | 1.37 | $2.4 \times 10^7$ | $2.5 \times 10^7$ | 5.13 |
| $k_{d2}$ | $2.4 \times 10^7$ | $2.5 \times 10^7$ | 2.66 | $2.4 \times 10^7$ | $2.5 \times 10^7$ | 3.15 | $2.0 \times 10^6$ | $2.1 \times 10^6$ | 5.83 |
| $k_{d3}$ | $2.4 \times 10^7$ | $2.6 \times 10^7$ | 6.83 | $2.0 \times 10^6$ | $2.0 \times 10^6$ | 2.33 | $2.0 \times 10^6$ | $2.1 \times 10^6$ | 3.14 |
| $k_{d4}$ | $3.0 \times 10^8$ | $3.1 \times 10^8$ | 4.18 | $2.5 \times 10^7$ | $2.6 \times 10^7$ | 5.67 | $2.5 \times 10^7$ | $2.6 \times 10^7$ | 2.25 |
| $k_{d5}$ | $2.5 \times 10^7$ | $2.6 \times 10^7$ | 2.35 | $3.0 \times 10^8$ | $3.1 \times 10^8$ | 4.35 | $3.0 \times 10^8$ | $3.1 \times 10^8$ | 3.64 |
| $k_{d6}$ | $1.0 \times 10^8$ | $1.1 \times 10^8$ | 7.24 | $3.0 \times 10^8$ | $3.1 \times 10^8$ | 3.15 | $1.9 \times 10^8$ | $1.9 \times 10^8$ | 1.29 |
| $k_{d7}$ | $9.4 \times 10^6$ | $9.6 \times 10^6$ | 1.37 | $1.1 \times 10^7$ | $1.1 \times 10^7$ | 1.23 | $9.3 \times 10^5$ | $9.5 \times 10^5$ | 1.55 |
| $k_{d8}$ | $1.7 \times 10^7$ | $1.7 \times 10^7$ | 2.14 | $6.3 \times 10^6$ | $6.4 \times 10^6$ | 0.87 | $2.4 \times 10^6$ | $2.4 \times 10^6$ | 0.56 |

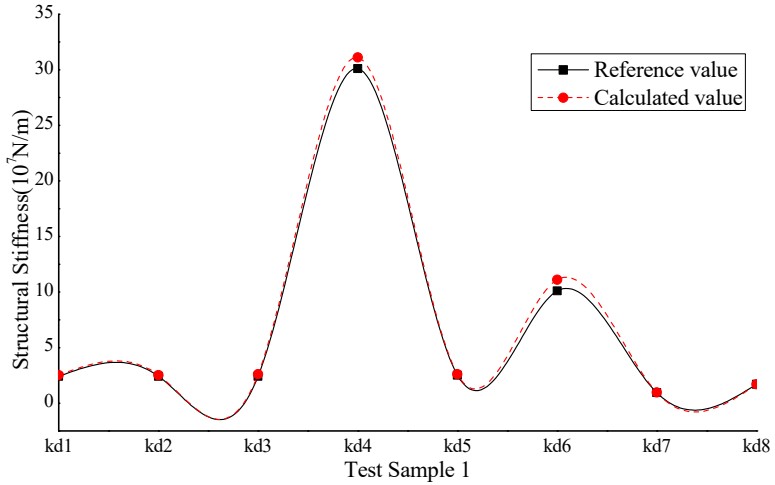

Test results of sample 1

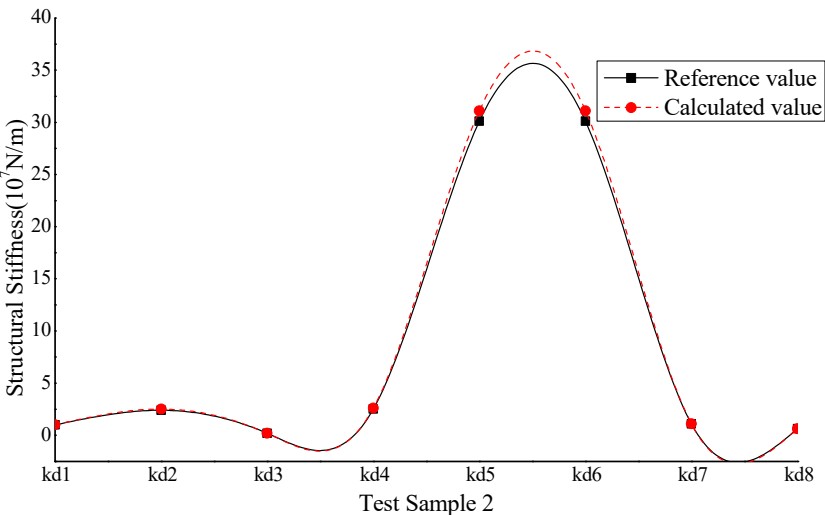

Test results of sample 2

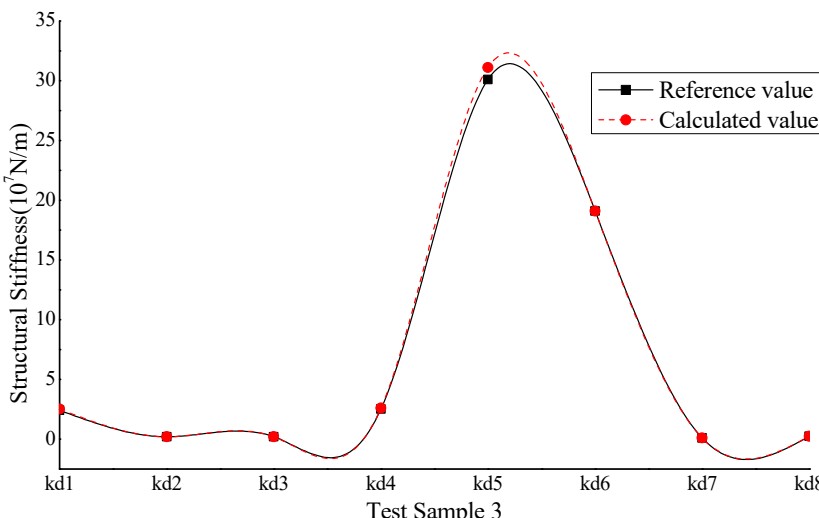

Test results of sample 3

**Figure 8.** Damage identification results of test samples.

## 4.3. Seismic Damage Identification Based on WNSVM

Based on the WNSVM which had been trained and tested, the measured accelerations of the 1:30 scale composite cable-stayed bridge model was used to seismic damage identification in each case. The measured samples are shown in Table 6, and the identified results have been shown graphically in Figure 8.

**Table 6.** Measured samples.

| Case | 1 | 2 | 3 | 4 | 5 | 6 | 7 | 8 | 9 | 10 |
|------|-----|-----|-----|-----|-----|-----|-----|-----|-----|-----|
| $k_1$ | $2.0 \times 10^7$ | $2.0 \times 10^7$ | $2.0 \times 10^7$ | $2.0 \times 10^7$ | $2.0 \times 10^7$ | $2.0 \times 10^7$ | $2.0 \times 10^7$ | $2.0 \times 10^7$ | $2.0 \times 10^7$ | $2.0 \times 10^7$ |
| $k_2$ | $2.0 \times 10^7$ | $2.0 \times 10^7$ | $2.0 \times 10^7$ | $2.0 \times 10^7$ | $2.0 \times 10^7$ | $2.0 \times 10^7$ | $2.0 \times 10^7$ | $2.0 \times 10^7$ | $2.0 \times 10^7$ | $2.0 \times 10^7$ |
| $k_3$ | $2.0 \times 10^7$ | $2.0 \times 10^7$ | $2.0 \times 10^7$ | $2.0 \times 10^7$ | $2.0 \times 10^7$ | $2.0 \times 10^7$ | $2.0 \times 10^7$ | $2.0 \times 10^7$ | $2.0 \times 10^7$ | $2.0 \times 10^7$ |
| $k_4$ | $2.5 \times 10^7$ | $2.5 \times 10^7$ | $2.5 \times 10^7$ | $2.5 \times 10^7$ | $2.5 \times 10^7$ | $2.5 \times 10^7$ | $2.5 \times 10^7$ | $2.5 \times 10^7$ | $2.5 \times 10^7$ | $2.5 \times 10^7$ |
| $k_5$ | $2.5 \times 10^7$ | $2.5 \times 10^7$ | $2.5 \times 10^7$ | $2.5 \times 10^7$ | $2.5 \times 10^7$ | $2.5 \times 10^7$ | $2.5 \times 10^7$ | $2.5 \times 10^7$ | $2.5 \times 10^7$ | $2.5 \times 10^7$ |
| $k_6$ | $2.5 \times 10^7$ | $2.5 \times 10^7$ | $2.5 \times 10^7$ | $2.5 \times 10^7$ | $2.5 \times 10^7$ | $2.5 \times 10^7$ | $2.5 \times 10^7$ | $2.5 \times 10^7$ | $2.5 \times 10^7$ | $2.5 \times 10^7$ |
| $k_7$ | $9.3 \times 10^6$ | $9.3 \times 10^6$ | $9.3 \times 10^6$ | $9.3 \times 10^6$ | $9.3 \times 10^6$ | $9.3 \times 10^6$ | $9.3 \times 10^6$ | $9.3 \times 10^6$ | $9.3 \times 10^6$ | $9.3 \times 10^6$ |
| $k_8$ | $2.4 \times 10^7$ | $2.4 \times 10^7$ | $2.4 \times 10^7$ | $2.4 \times 10^7$ | $2.4 \times 10^7$ | $2.4 \times 10^7$ | $2.4 \times 10^7$ | $2.4 \times 10^7$ | $2.4 \times 10^7$ | $2.4 \times 10^7$ |
| $a_1$ | 0.43 | 0.34 | 0.31 | 0.40 | 0.64 | 1.19 | 1.58 | 1.91 | 2.47 | 3.04 |
| $a_2$ | 0.43 | 0.34 | 0.31 | 0.40 | 0.63 | 1.18 | 1.54 | 1.89 | 2.57 | 3.20 |
| $a_3$ | 0.32 | 0.25 | 0.23 | 0.30 | 0.51 | 0.95 | 1.14 | 1.48 | 1.79 | 2.45 |
| $a_4$ | 0.21 | 0.26 | 0.08 | 0.11 | 0.49 | 0.31 | 0.42 | 1.47 | 0.63 | 0.94 |
| $a_5$ | 0.21 | 0.26 | 0.08 | 0.11 | 0.49 | 0.31 | 0.44 | 1.47 | 0.67 | 0.88 |
| $a_6$ | 0.18 | 0.22 | 0.07 | 0.10 | 0.42 | 0.27 | 0.36 | 1.25 | 0.54 | 0.73 |
| $a_7$ | 0.62 | 0.48 | 0.45 | 0.58 | 0.96 | 1.70 | 2.19 | 2.87 | 3.46 | 4.76 |
| $a_8$ | 0.98 | 1.21 | 0.38 | 0.52 | 2.29 | 1.51 | 2.06 | 6.80 | 2.93 | 4.40 |
| $a_{d1}$ | 0.44 | 0.35 | 0.32 | 0.41 | 0.83 | 1.89 | 2.36 | 3.42 | 7.67 | 15.64 |
| $a_{d2}$ | 0.45 | 0.36 | 0.33 | 0.43 | 0.71 | 1.34 | 2.00 | 2.85 | 9.15 | 13.61 |
| $a_{d3}$ | 0.34 | 0.25 | 0.24 | 0.32 | 0.60 | 1.35 | 2.08 | 2.56 | 10.13 | 21.45 |
| $a_{d4}$ | 0.21 | 0.27 | 0.09 | 0.12 | 0.54 | 0.34 | 0.47 | 1.97 | 6.39 | 9.33 |
| $a_{d5}$ | 0.20 | 0.26 | 0.09 | 0.11 | 0.54 | 0.35 | 0.45 | 2.18 | 1.27 | 2.90 |
| $a_{d6}$ | 0.18 | 0.23 | 0.07 | 0.10 | 0.47 | 0.30 | 0.40 | 1.61 | 2.34 | 13.39 |
| $a_{d7}$ | 0.65 | 0.51 | 0.46 | 0.62 | 0.98 | 1.93 | 2.48 | 2.93 | 4.16 | 5.44 |
| $a_{d8}$ | 0.97 | 1.18 | 0.40 | 0.51 | 2.54 | 1.54 | 2.10 | 7.69 | 4.20 | 4.96 |

Compared the identified results in Figure 9 with the test results in Table 7, it is clear that: (1) After the structure was subjected to earthquake excitation of case 5, the stiffness of the support in X direction decreased, which is consistent with the phenomenon that minor cracks occur in the support in X direction found in the shaking table tests; (2) Under the earthquake excitation of cases 6–8, the stiffness of the support in the X direction is further reduced, and a large number of cracks are found in the supports during the tests; (3) Under the earthquake excitation of cases 9 and 10, the stiffness of supports and the bridge tower are decreased significantly, of which the supports stiffness decreased about 85%. In the experiment, lots of cracks appeared on the surfaces of supports, stay cable anchor block, and bridge tower. It is clear that the WNSVM method can be used for the seismic damage identification of composite cable-stayed bridges.

**Table 7.** Damage identification based on WNSVM.

| Case | Seismic Waves | Damage Descriptions | X-Direction Support | Y-Direction Support | k7 (N/m) Y-Direction Bridge Tower | k8 (N/m) X-Direction Bridge Tower |
|------|---------------|---------------------|---------------------|---------------------|-----------------------------------|-----------------------------------|
| | | | (d1 + d2 + d3)/3 | (d4 + d5 + d6)/3 | d7 | d8 |
| 1 | Lander-amboy | No obvious cracks | 0.96 | 1.02 | 0.95 | 1.01 |
| 2 | Cerro Prieto | No obvious cracks | 0.96 | 0.96 | 0.94 | 1.02 |
| 3 | Chi-chi | No obvious cracks | 0.97 | 0.95 | 0.99 | 0.95 |

**Table 7.** *Cont.*

| Case | Seismic Waves | Damage Descriptions | X-Direction Support | Y-Direction Support | k7 (N/m) Y-Direction Bridge Tower | k8 (N/m) X-Direction Bridge Tower |
|---|---|---|---|---|---|---|
| | | | (d1 + d2 + d3)/3 | (d4 + d5 + d6)/3 | d7 | d8 |
| 4 | El-Centro | No obvious cracks | 0.96 | 0.96 | 0.94 | 1.02 |
| 5 | Cerro Prieto | Minor cracks in X-direction support | 0.86 | 0.95 | 0.99 | 0.95 |
| 6 | El-Centro | Lots of cracks in X-direction support | 0.77 | 0.95 | 0.94 | 0.99 |
| 7 | Chi-chi | Lots of cracks in X-direction support | 0.69 | 0.96 | 0.94 | 0.99 |
| 8 | El-Centro | Lots of cracks in X-direction support, Minor cracks in Y-direction support and stay cable anchor block | 0.63 | 0.78 | 0.99 | 0.94 |
| 9 | Chi-chi | Lots of cracks in X-direction and Y-direction support, Minor cracks in stay cable anchor block and bridge tower | 0.25 | 0.29 | 0.84 | 0.74 |
| 10 | Chi-chi | lots of major cracks appeared | 0.19 | 0.15 | 0.86 | 0.84 |

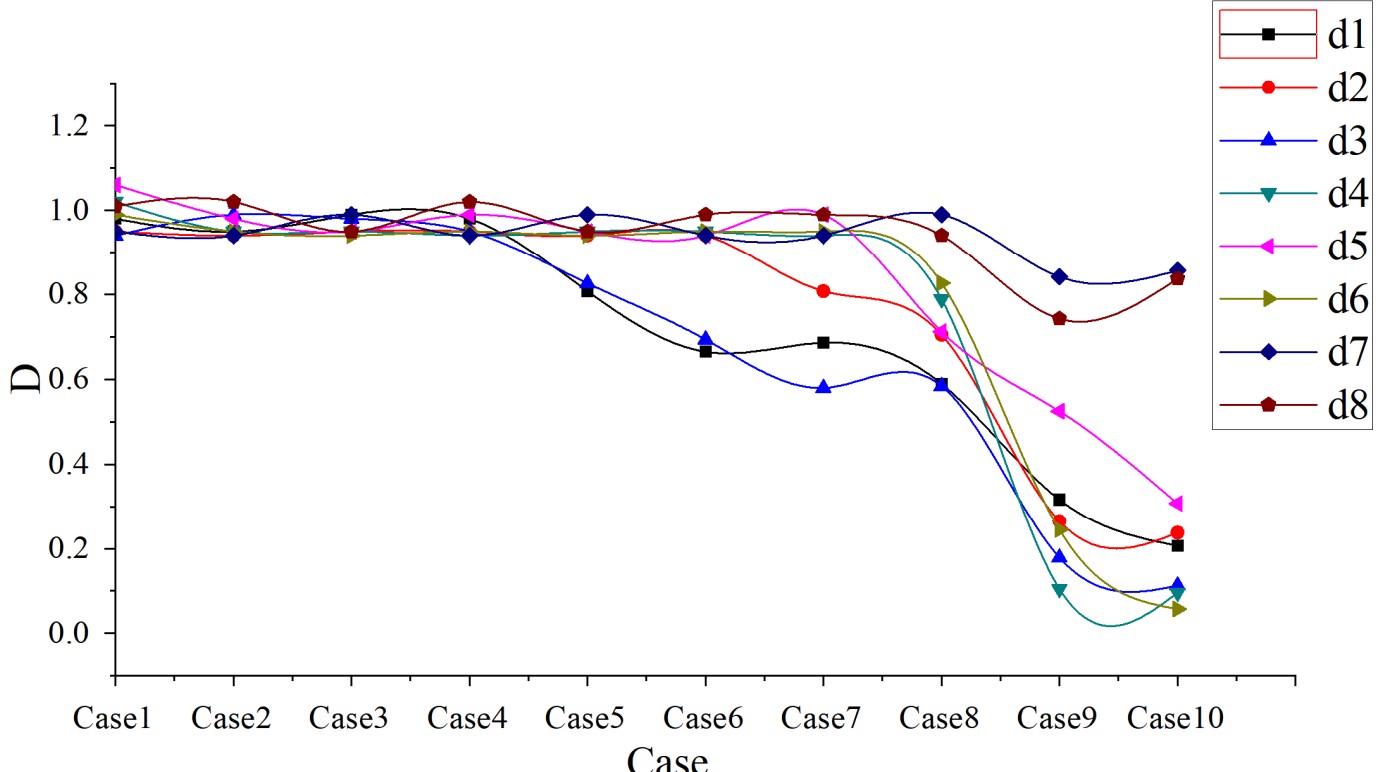

**Figure 9.** Damage identification based on WNSVM.

## 5. Conclusions

In this study, a method for seismic damage identification, which is intended for composite cable-stayed bridges, has been developed. A shaking table test is employed to demonstrate the implementation and potential applications of the proposed method for seismic damage identification. The following conclusions can be drawn from this study:

(1). The structure responses of single-tower cable-stayed bridge under earthquake excitations can be calculated based on the nonlinear FEM, the nonlinear FEM can correctly simulate the single-tower cable-stayed bridge model, and the learning samples of the WNSVM model in this study can be produced based on the nonlinear FEM.

(2). The structural damage results identified by the WNSVM method are in good agreement with those obtained by the shaking table test, and the maximum error is less than 8%. Therefore, the WNSVM method can be used for the seismic damage identification of composite cable-stayed bridges.

**Author Contributions:** Methodology, Z.S. and R.Z.; Software, Z.S.; Validation, R.Z.; Formal analysis, Z.S.; Resources, R.Z.; Data curation, N.J.; Writing—original draft, N.J. All authors have read and agreed to the published version of the manuscript.

**Funding:** This research was funded by [National Natural Science Foundation of China] grant number [No. 52192664].

**Institutional Review Board Statement:** Not applicable.

**Informed Consent Statement:** Not applicable.

**Data Availability Statement:** Data is contained within the article.

**Acknowledgments:** The authors gratefully acknowledge the financial support provided by the National Natural Science Foundation of China (No. 52192664). The viewpoints of this paper represent only the author's opinion; they do not represent the views of the fund committee.

**Conflicts of Interest:** The authors declare no conflict of interest.

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
