# Peer review of "Seismic Damage Identification of Composite Cable-Stayed Bridges Using Support Vector Machines and Wavelet Networks"

_sustainability, doi:10.3390/su15010108_

Round 1
Reviewer 1 Report
1. Kindly mention the details of the bridge under consideration. Composite is a generic term and has to be specified with exact details.
2. Kindly rearrange the sections 2 and 3 in a better sequence by either flipping them or developing a combined section starting with the experimental details.
3. There could be plenty full of the information required regarding the model used. Kindly mention all necessary details namely materials used, joint types with pictures, connection details with pictures, illustrative figure for the experimental setup as a whole and finally the simulation datasets utilized in computerized modeling process. These will fulfill the readers' needs better.
4. Again, there is a mix up of experimental description and results obtained in the section 3. Kindly make it more specific, independently described and systematically narrated.
5. Fig. 9 needs detailed explanation about the specific trends obtained. Relate the results with the predicted or expected response and also with the proposed methodology by a comparison table (recommended).
6. The conclusion is in the form of discussion to an extent. Kindly make sure that in this part, only author's own scientifically cross checked and confirmed statements are written. Generally, each statement of a conclusion section should stand alone and though in the good coherence with the aims and objectives of the work. Kindly rework accordingly.
Reviewer 3 Report
The manuscript “Seismic damage identification of composite cable-stayed bridges using support vector machines and wavelet networks”, noted as sustainability-2083627, adopted wavelet networks (WN) and support vector machines (SVM) methods for the seismic damage identification of composite cable-stayed bridges. Experimental analysis and finite element analysis (FEA) were conducted to generate data for training and testing. Conclusions were drawn that the proposed method can offer a satisfactory evaluation of seismic damage location.
Overall, the research topic is interesting. However, I think this manuscript needs critical revision in several following aspects:
------------------------------------------------------------
1. In the shake table test, the earthquake excitations with different intensity scales were continuously applied to the bridge structure. How did the authors take the damage accumulation into consideration during this continuous loading process? There might be the possibility that one damage zone was accumulated and resulted from multiple excitations.
------------------------------------------------------------
2. Table/figure captions and figures’ x, y axes labels explanation. During the review, the reviewer found that the authors used very simple description of tables, figures, and figures x, y axes labels (e.g. Table 3, Figure 9), which was not very helpful for the readers to understand the information conveyed by those figures/tables. Please provide sufficient information in the captions and labels.
------------------------------------------------------------
3. The authors adopted two machine learning models SVM and WN in their methodology. However, by reading the paper, the reviewer found that there were no clear and concise summary of what’s input (ground motion intensity measures, structural parameters?) and output (damage index?) of these two machine learning models? What hyperparameters (learning rate, optimizer, etc.) were used for model training? Tables would be very helpful to summarize these critical information for machine learning model training and testing.
------------------------------------------------------------
4. The authors conducted extensive literature reviews of data-driven seismic damage identification. However, the authors used some publications which were more than a decade ago and omitted some latest progress in this domain. Several publications were listed below for the authors to consider for reference.
Yuan, X., Chen, G., Jiao, P., Li, L., Han, J., & Zhang, H. (2022). A neural network-based multivariate seismic classifier for simultaneous post-earthquake fragility estimation and damage classification. Engineering Structures, 255, 113918.
Zhong, Jian; Shi, Longfei, Yang, Tao; Liu, Xiaoxian*; Wang, Yixian. Probabilistic seismic demand model of UBPRC columns conditioned on Pulse-Structure parameters. Engineering Structures. 2022,270:114829.
Yang, Tao; Yuan, Xinzhe; Zhong, Jian, Yuan, Wancheng; Near-fault Pulse Seismic Ductility Spectra for Bridge Columns Based on Machine Learning. Soil Dynamics and Earthquake Engineering, 2023, 164: 107582.
Zhong, Jian; Ni, Ming; Hu, Huiming; Yuan, Wancheng; Yuan, Haiping; Pang, Yutao*; Uncoupled multivariate power models for estimating performance-based seismic damage states of column curvature ductility; Structures, 2022, 36: 752-764.
------------------------------------------------------------
5. Please screen the manuscript for written and syntax error. The reviewer suggested the authors consult professional English writers or proofreaders to polish the manuscript. Several examples of incorrect writings are:
Line 49: “Comparing” should be “Compared”
Line 82: “mode rate” should be “moderate”
Line 221: “seismic destroy” is confusing. “seismic damage” is recommended.
Line 272-273: this sentence is confusing.
------------------------------------------------------------
Round 2
Reviewer 3 Report
Overall, the authors correctly responded and addressed the reviewer's comments. Please further check the written English in the following proofreading phase as the review still observed some syntax errors like Line 284: Comparing should be Compared.
One more comment is Line 53: Zhong et al. [36-38] should be Zhong et al. [36, 38] and Yang et al. [37] systematically discussed the method...
Author Response
The authors thank the editor and the reviewers for valuable comments, some syntax errors have been revised in the new manuscript.